

# A comprehensive analysis identified an autophagy-related risk model for predicting recurrence and immunotherapy response in stage I lung adenocarcinoma

Hongmei Zheng[1,2], Songqing Fan[1,2], Hongjing Zang[1,2], Jiadi Luo[1,2], Long Shu[3] and Jinwu Peng[4]

[1] Department of Pathology, The Second Xiangya Hospital, Central South University, Changsha, China
[2] Hunan Clinical Medical Research Center for Cancer Pathogenic Genes Testing and Diagnosis, Changsha, Hunan, China
[3] Department of Oncology, Xiangya Hospital, Central South University, Changsha, China
[4] Department of Pathology, Xiangya Hospital, Central South University, Changsha, China

Corresponding authors
Long Shu, 1216854140@qq.com
Jinwu Peng, Jinwupeng2005@aliyun.com

## ABSTRACT

**Background:** Lung adenocarcinoma (LUAD) is characterized by early recurrence and poor prognosis. Autophagy is a double-edged sword in tumor development and anti-tumor therapy resistance. However, the prediction of relapse and therapeutic response in LUAD patients with stage I based on the signature of autophagy remains unclear.

**Methods:** Gene expression data were obtained from the Gene Expression Omnibus (GEO) and The Cancer Genome Atlas (TCGA) database. Autophagy-associated genes were extracted from the Human Autophagy Moderator Database. The autophagy score was established by Least Absolute Shrinkage and Selection Operator (LASSO) regression. Real-time PCR was used to detect gene expression of hub genes in LUAD patients. Protein-protein interaction (PPI) was analyzed to identify crucial genes. Gene set enrichment analysis (GSEA) was used to reveal the molecular features of patients. ESTIMATE algorithm was applied to estimate the tumor immune infiltration. TIDE score and Genomics of Drug Sensitivity in Cancer (GDSC) database were used to assess therapeutic response.

**Results:** We established an autophagy score based on 19 autophagy genes. Among these genes, MAP1LC3B played a crucial role in PPI network and was down-regulated in tumor tissues both in TCGA and local cohort. Receiver operating characteristic (ROC) curve showed that the risk model effectively predict RFS of stage I LUAD (area under the curve (AUC) at 1, 2, 3 years = 0.701, 0.836, and 0.818, respectively). Multivariate regression analysis indicated that the autophagy score was an independent predictor for relapse ($P < 0.001$, HR = 4.8, 95% CI [3.25–7.2]). The autophagy score also showed great predictive efficacy in the external validation GEO cohorts. GSEA revealed gene sets significantly enriched in immunity, cell cycle, and adhesion, *etc.* Meanwhile, we found the autophagy score was negatively related to KRAS mutation ($P = 0.017$) but positively associated with TP53 mutation ($P = 6.4e−11$). The autophagy score had a negative relationship with CD8+, CD4+ T cell, and dendritic cell, and positively correlated with immune checkpoint molecule CD276.
Patients with a high autophagy score were sensitive to chemotherapy and targeted therapy, while resistant to immune checkpoint inhibitors.

**Conclusion:** We constructed an effective recurrence risk predictive model for stage I LUAD patients based on autophagy related genes. High autophagy score predicted a higher recurrence risk and suppressing tumor immune microenvironment.

## INTRODUCTION

A total of 85% of lung cancer is non-small cell lung cancer (NSCLC), which is one of the highest incidences and mortality rates (*Siegel et al., 2022*; *Sung et al., 2021*). Among them, the most frequent histological type is LUAD, which is characterized by early recurrence and poor prognosis (*Perez-Johnston et al., 2022*; *Xia et al., 2022*). Although these patients' prognosis has been improved with the progress of chemotherapy, targeted therapy and immunotherapy, the overall survival (OS) is still low because of early recurrence and drug resistance (*Herbst, Morgensztern & Boshoff, 2018*). Hence, it is particularly important to timely identify patients who are prone to relapse and give them personalized treatment strategies.

As one of the cell catabolism ways, autophagy degrades potentially harmful substances through lysosomes, such as damaged organelles, pathogenic microorganisms and misfolded proteins, which plays a role in protecting the homeostasis of the intracellular environment (*Russell & Guan, 2022*). Autophagy has been proved to be related to the pathogenesis of many human diseases, such as neurodegenerative disease and tumor (*Zhang et al., 2022*; *Fan et al., 2017*; *Singh et al., 2018*; *Gao et al., 2022*). Autophagy plays a different role in tumor development, excessive autophagy cause cell death at early stage. In the middle and late stages of tumor development, autophagy generally increases, which promotes the survival of tumor cells and increases the malignancy of tumors (*Poillet-Perez & White, 2019*). In addition, autophagy may also act as a dual role in anti-tumor therapy resistance (*Russell & Guan, 2022*; *Singh et al., 2018*). More and more studies are about the roles of autophagy-related signature in different cancers. A study has reported the autophagy-based signature could well predict the LUAD patients' prognosis and their immunotherapy efficiency (*Li et al., 2022*). *Liu et al. (2019)* also reported the prognostic significance of autophagy-related signature in NSCLC. In addition, *Fu et al. (2021)* identified an autophagy-associated signature can predict immune microenvironment feature and prognosis in acute myeloid leukemia. Autophagy-related gene signatures can also well predict the prognosis in glioblastoma and hepatocellular carcinoma (*Wang et al., 2019*; *Fang & Chen, 2020*). However, whether autophagy related signature may become biomarkers of tumor relapse and therapeutic response in stage I LUAD patients is still a knowledge gap.

In this research, multiple databases, such as The Cancer Genoma Atlas (TCGA), Gene Expression Omnibus (GEO) data and a series of bioinformatic methods were used to

**Table 1 Clinicopathological characteristics of patients with stage I LUAD in TCGA cohort and GEO cohort.**

|  | TCGA (*N* = 269) | GSE30219 (*N* = 71) | GSE37745 (*N* = 34) | Overall (*N* = 374) |
|---|---|---|---|---|
| Age (years) |  |  |  |  |
| Average | 65.9 | 61.2 | 63.8 | 64.8 |
| Median | 67.0 | 60.0 | 66.0 | 65.0 |
| Unknown | 7 | 0 | 0 | 7 |
| Gender |  |  |  |  |
| Female | 158 (58.7%) | 16 (22.5%) | 21 (61.8%) | 195 (52.1%) |
| Male | 111 (41.3%) | 55 (77.5%) | 13 (38.2%) | 179 (47.9%) |
| Smoking history |  |  |  |  |
| No | 37 (13.8%) | 0 (0%) | 0 (0%) | 37 (9.9%) |
| Yes | 228 (84.8%) | 0 (0%) | 0 (0%) | 228 (61.0%) |
| Unknown | 4 (1.4%) | 71 (100%) | 34 (100%) | 109 (29.1%) |
| RFS (days) |  |  |  |  |
| Average | 856 | 2,180 | 1,880 | 1,200 |
| Median | 578 | 1,920 | 1,730 | 652 |
| Recurrence |  |  |  |  |
| No | 182 (67.7%) | 54 (76.1%) | 19 (55.9%) | 255 (68.2%) |
| Yes | 87 (32.3%) | 17 (23.9%) | 15 (44.1%) | 119 (31.8%) |

Note:
LUAD, Lung Adenocarcinoma; RFS, Relapse Free Survival.

construct and validate a novel autophagy-related risk model to predict the early relapse of stage I LUAD patients. In addition, we explored the clinical characteristics and the immune landscape of this model. Eventually, the potential value of our model in predicting therapeutic response has also been examined. Therefore, this study will help to promote individualized treatment and thereby may reduce the relapse rate of LUAD patients with stage I.

## MATERIALS AND METHODS

### Data source collection

In total, we obtained 269 stage I LUAD data with relevant recurrence-free survival (RFS) and clinical features from TCGA, downloaded from the Cancer Genomics Browser of The University of California Santa Cruz (UCSC) (http://xena.ucsc.edu/). Furthermore, all autophagy related genes were derived from the human autophagy moderator database (HAMdb, http://hamdb.scbdd.com/). TCGA data was assigned as a training cohort, whereas GEO datasets (http://www.ncbi.nlm.nih.gov/geo) were used as the external validation cohorts (GSE30219 and GSE37745). All data were shown in Table 1.

### Risk model construction and validation analysis

A univariate Cox proportional analysis was used to investigate the association between autophagy genes and RFS with $P < 0.05$. Next, we attempted to construct autophagy-related genes based signatures for recurrence prediction by Least Absolute

Shrinkage and Selection Operator (LASSO) regression analysis and construct a risk model containing mRNA expression levels and coefficients, named autophagy score. The LUAD patients were divided into the high-risk and low-risk group based on the cut-off value of the Kaplan-Meier curve. Decision curve analysis (DCA), receiver operating characteristic (ROC) analysis, and multivariate Cox regression were applied to test the stability and suitability of this model. Finally, GEO datasets were similarly analyzed to further verify the significance of the model.

## Biological process and pathway enrichment analysis

Gene Ontology (GO) enrichment analysis and Kyoto Encyclopedia of Genes and Genomes (KEGG) pathway were performed using the R GSVA package. $P$ value < 0.05 and an absolute value of normalized enrichment score (NES) value >1 was considered as significant enrichment.

## Tumor immune microenvironment and immune checkpoint profile analysis

The ESTIMATE algorithm (*Yoshihara et al., 2013*) was applied to estimate the proportion and distribution of tumor-infiltrating immune cells. Both gene mutation status and immune checkpoint molecules were used to explore the association with the autophagy score.

## Risk score response to chemotherapy, targeted therapy, and immunotherapy

Chemotherapy and targeted therapy drugs response prediction for each sample were evaluated by the oncoPredict R package. The tumor immune dysfunction and exclusion (TIDE) (*Jiang et al., 2018*) score has been applied as a biomarker of immunotherapy response.

## Ethics approval and consent to participate

This study was reviewed and approved by The Ethics Review Committee of Central South University, The Second Xiangya Hospital (LYEC2024-0253). Written consent was obtained from the patients.

## Detect the mRNA expression levels of hub genes of the autophagy score in normal lung tissue and LUAD tissues by qPCR

Fresh normal lung tissues and tumor tissues from LUAD patients were obtained from the Second Xiangya Hospital. The mRNA expression level of hub genes of the autophagy score were assessed by qPCR. The primer sequences used in this study were provided in the Table 2.

## Statistical analysis

All statistical analyses were performed on R software. The Wilcoxon test was used for comparison between the two groups, and Spearman test was used for all correlation analyses. $P < 0.05$ was considered significant.

**Table 2 The primer sequences used in this study.**

| Gene name | Forward sequence | Reverse sequence |
|---|---|---|
| CALCOCO2 | CCAGTTCTGCTATGTGGATGAGG | GTGCTGCTCAATCTCTTCCACC |
| MAP1LC3B | GAGAAGCAGCTTCCTGTTCTGG | GTGTCCGTTCACCAACAGGAAG |
| TRIM8 | GACGTGGAGATCCGAAGGAATG | CAGCCGAACTTCCTCCTTCAGT |
| PARK2 | CCAGAGGAAAGTCACCTGCGAA | CTGAGGCTTCAAATACGGCACTG |
| MAPT | CCAGTCCAAGTGTGGCTCAAAG | GCCTAATGAGCCACACTTGGAG |
| PDK2 | TGCCTACGACATGGCTAAGCTC | GACGTAGACCATGTGAATCGGC |
| PTEN | TGAGTTCCCTCAGCCGTTACCT | GAGGTTTCCTCTGGTCCTGGTA |
| GABBR2 | GTTGCTCAAGCACTACCAGTGG | TCCTCGCCATACAGAACTCCAG |
| CXCR3 | ACGAGAGTGACTCGTGCTGTAC | GCAGAAAGAGGAGGCTGTAGAG |
| TFEB | CCTGGAGATGACCAACAAGCAG | TAGGCAGCTCCTGCTTCACCAC |
| GPR37 | ATGTCGCGGCTACTGCTTC | GCAGAACGTCTCTTGCAGAAT |
| RPS6KB1 | TATTGGCAGCCCACGAACACCT | GTCACATCCATCTGCTCTATGCC |
| GRID2 | TCTTACACGGCAAACCTCGCTG | TACCGCAGAGTCTAGGACTGTG |
| VPS37D | ATACCAGGAGCTTCGTGAGGTG | CCTCTTCTAGCTCAGCCTGCAG |
| CHMP4B | ACCAACACCGAGGTGCTCAAGA | CTGCAAGTTCTTGCTGGTCAGC |
| GOPC | AGAAGGAGGTGGTAACCCTGGT | TTGAAGCACCGTCATCTAGCGG |
| PIK3CA | GAAGCACCTGAATAGGCAAGTCG | GAGCATCCATGAAATCTGGTCGC |
| BCL2L1 | GCCACTTACCTGAATGACCACC | AACCAGCGGTTGAAGCGTTCCT |
| RAB1A | GGGAACAAATGTGATCTGACCAC | GAAAGACTGTTCTACATTCGTTGC |

# RESULTS

## Identification of an autophagy-related risk model for the recurrence of LUAD

Based on 269 LUAD patients with stage I from the TCGA database, we conducted univariate Cox regression and discovered 43 autophagy genes that significantly related to cancer recurrence. The top 10 genes were shown in Fig. 1A, such as RAB1A, MFN1 and XPO1. Then, we performed LASSO regression analysis (Figs. 1B and 1C) and identified 19 related hub genes (Fig. 1D) related to the recurrence of LUAD patients. By integrating these 19 genes, we developed the risk model, named as the autophagy score, including 10 protective factors (CALCOCO2, MAP1LC3B, TRIM8, PARK2, MAPT, PDK2, PTEN, GABBR2, CXCR3 and TFEB) and nine risk factors (RAB1A, BCL2L1, PIK3CA, GOPC, CHMP4B, VPS37D, GRID2, RPS6KB1 and GPR37). The risk score was as follows: autophagy score = 0.4563488 * RAB1A + 0.318107141 * BCL2L1 + 0.253867908 * PIK3CA + 0.237767088 * GOPC + 0.18501832 * CHMP4B + 0.114654237 * VPS37D + 0.10932385 * GRID2 + 0.026645972 * RPS6KB1 + 0.00210073 * GPR37 − 0.182756828 * CALCOCO2 − 0.147361444 * MAP1LC3B − 0.133470853 * TRIM8 − 0.105113941 * PARK2 − 0.073555698 * MAPT − 0.067317524 * PDK2 − 0.060281667 * PTEN − 0.053339503 * GABBR2 − 0.045620769 * CXCR3 − 0.026292117 * TFEB. In order to explore whether there was interaction between these hub genes, we applied STRING database to created

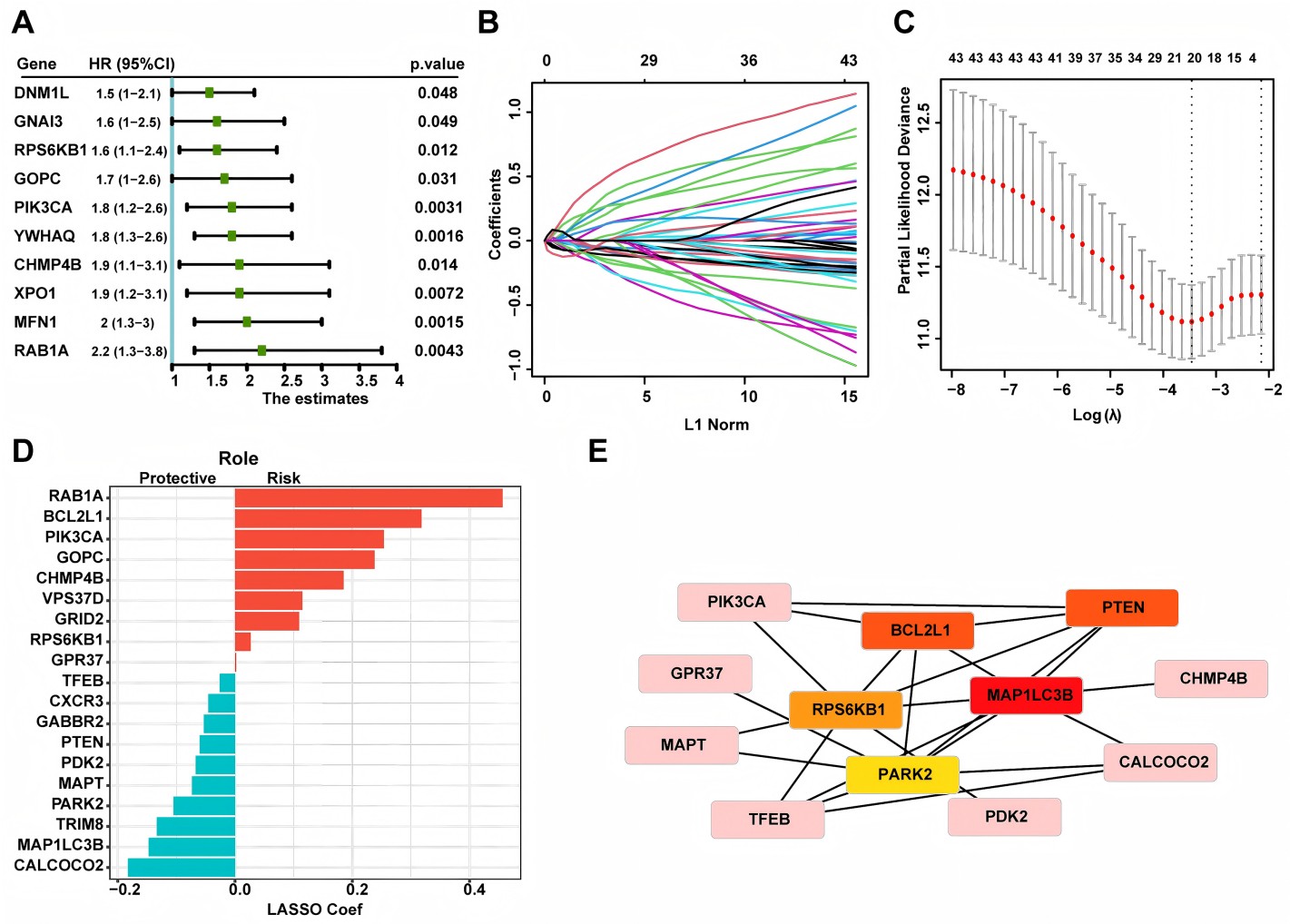

**Figure 1 Constructing autophagy-associated risk model to predict the recurrence in LUAD patients from TCGA database.** (A) The top 10 genes in univariate Cox regression. (B, C) LASSO analysis and cross validation identified 19 hub recurrence-associated genes. (D) 19 autophagy-associated hub genes and their coefficients. (E) PPI core network. LUAD, Lung Adenocarcinoma; PPI, Protein-Protein Interaction; LASSO, Least Absolute Shrinkage and Selection Operator.

PPI network. According to node degree, we identified MAP1LC3B was the core gene (Fig. 1E). Next, in the TCGA training cohort, patients were divided into two groups according to the above autophagy score, including low-risk and high-risk group. In addition, Kaplan-Meier curve analysis indicated that high-risk patients had a higher recurrence rate than low-risk patients (Fig. 2A, $P < 0.0001$). ROC curve analysis showed that AUC (area under the curve) of 1, 3 and 5 years were 0.701, 0.836 and 0.818 (Fig. 2B). The decision curve analysis for the autophagy score was presented in Fig. 2C. Moreover, multivariate Cox regression demonstrated the autophagy score was an independent predictor for RFS (Fig. 2D, HR = 4.8, 95% CI [3.25–7.2], $P < 0.001$). Also, we created a nomogram with four variables (gender, age, smoking history and autophagy score) to predict the patients' recurrence rate (Fig. 2E).

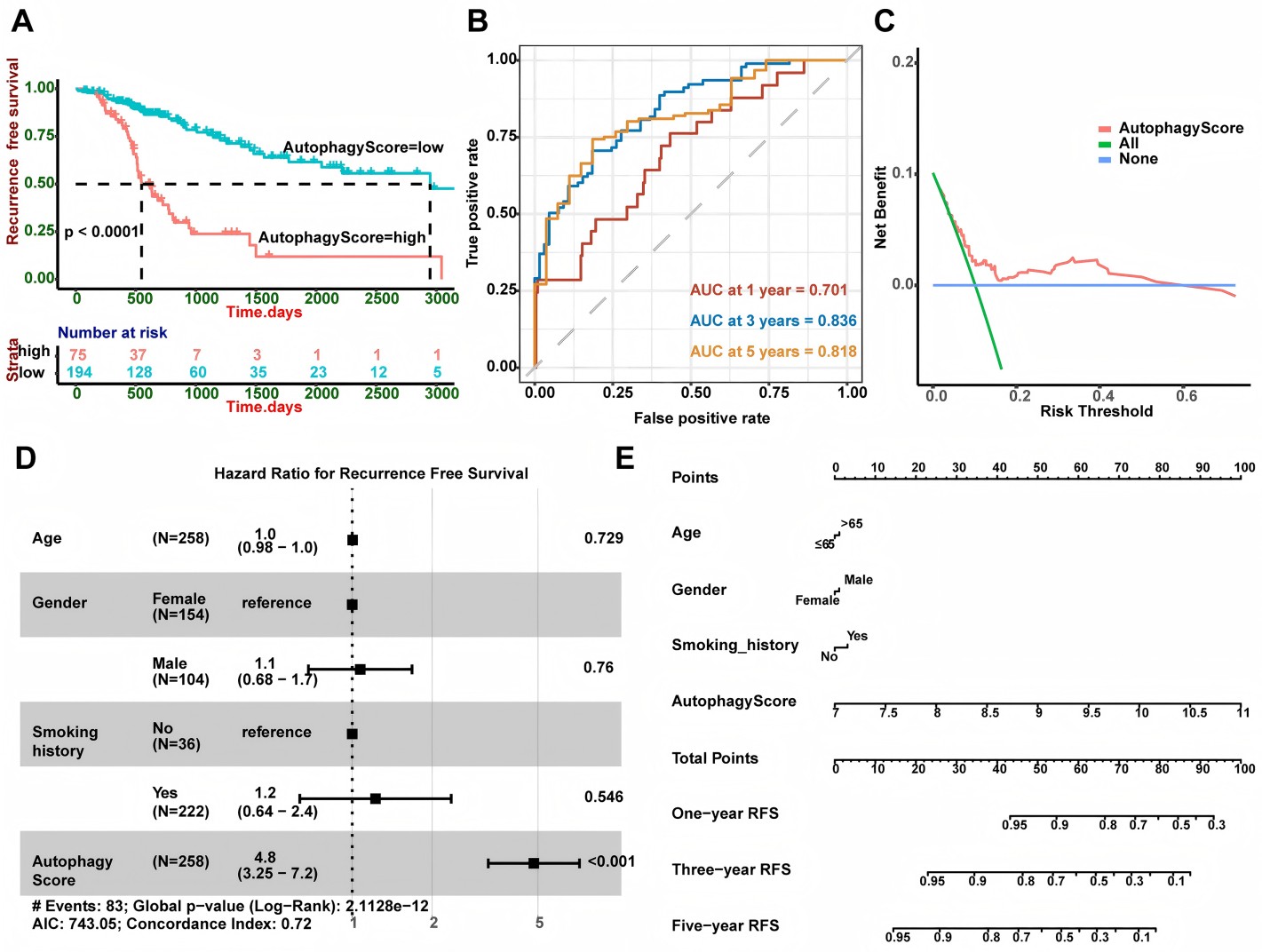

**Figure 2 Autophagy score could predict RFS.** (A) High autophagy score patients showed shorter RFS ($P < 0.0001$). (B) AUC at 1, 3, and 5 years of the autophagy score for RFS. (C) Decision curve analysis for the autophagy score. (D) Multivariate Cox regression showed the autophagy score was an independent predictor for RFS ($P < 0.001$). (E) The nomogram for calculating RFS in LUAD patients. LUAD, Lung A denocarcinoma; RFS, Recurrence-Free Survival; AUC, Area Under Curve.

## External validation of the risk model

GSE30219 and GSE37745 were utilized to verify the performance of our risk model. The survival analysis (Fig. 3) demonstrated that high-risk patients showed a worse RFS in GSE30219 ($P = 0.023$) and GSE37745 ($P = 0.0012$). A higher AUC value suggests better discrimination ability of the model, meaning it can more accurately distinguish between high- and low-risk groups. A value closer to 1 indicates excellent performance, while a value closer to 0.5 suggests a random model with no predictive power. So we performed ROC curve analysis and found that AUC of 1, 3 and 5 years were 0.541, 0.694 and 0.706 in GSE30219 and 0.656, 0.685 and 0.673 in GSE37745, which indicated the good performance of the risk model.
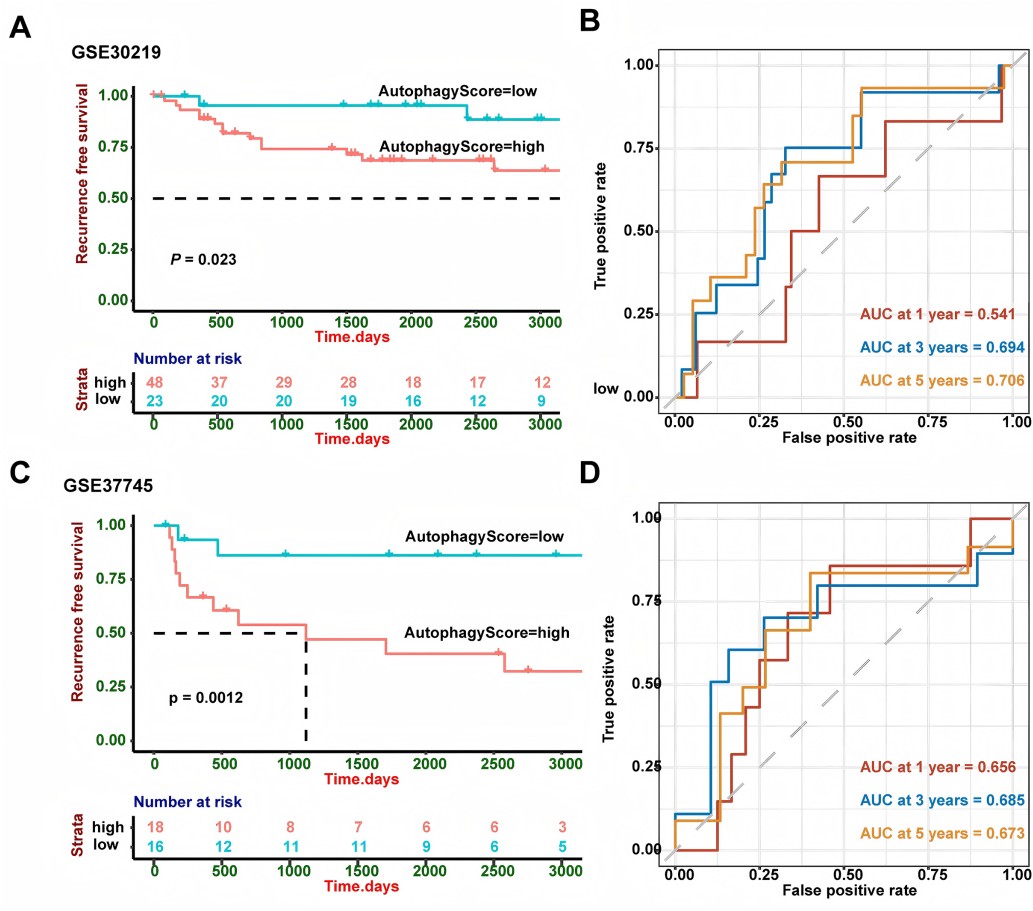

**Figure 3 External validation in different GEO cohorts.** (A) In GSE30219, Kaplan-Meier curve analysis showed patients with high-risk group had worse RFS ($P = 0.023$). (B) ROC curve analysis in GSE30219 cohort. (C) In GSE37745, Kaplan-Meier curve analysis showed patients with high-risk group had worse RFS ($P = 0.0012$). (D) ROC curve analysis in GSE37745 cohort. ROC, Receiver Operating Characteristic Curve; GEO, Gene Expression Omnibus; RFS, Recurrence-Free Survival.

## Biological pathway and functional enrichment analysis

Considering the satisfactory predictive value of the autophagy score in patients with LUAD, we further explored its potential mechanisms in the TCGA cohort by GSEA analysis (Fig. 4). The top three up-regulated GO BP terms were those involving the proteins that are involved in these pathways: cell cycle G2M phase transition, cell cycle G1S phase transition and chromosome segregation, whereas the top three down-regulated GO BP terms T cell activation, adaptive immune response and regulation of cell-cell adhesion. In addition, the top three up-regulated GO MF terms were those involving the proteins that are involved in these pathways: single stranded DNA binding, tubulin binding and catalytic activity acting on DNA and the down-regulated GO MF terms were cytokine receptor activity, passive transmembrane transporter activity, immune receptor activity and gated channel activity. Also, the top three up-regulated GO CC terms were those involving the proteins that are involved in these pathways: mitochondrial protein

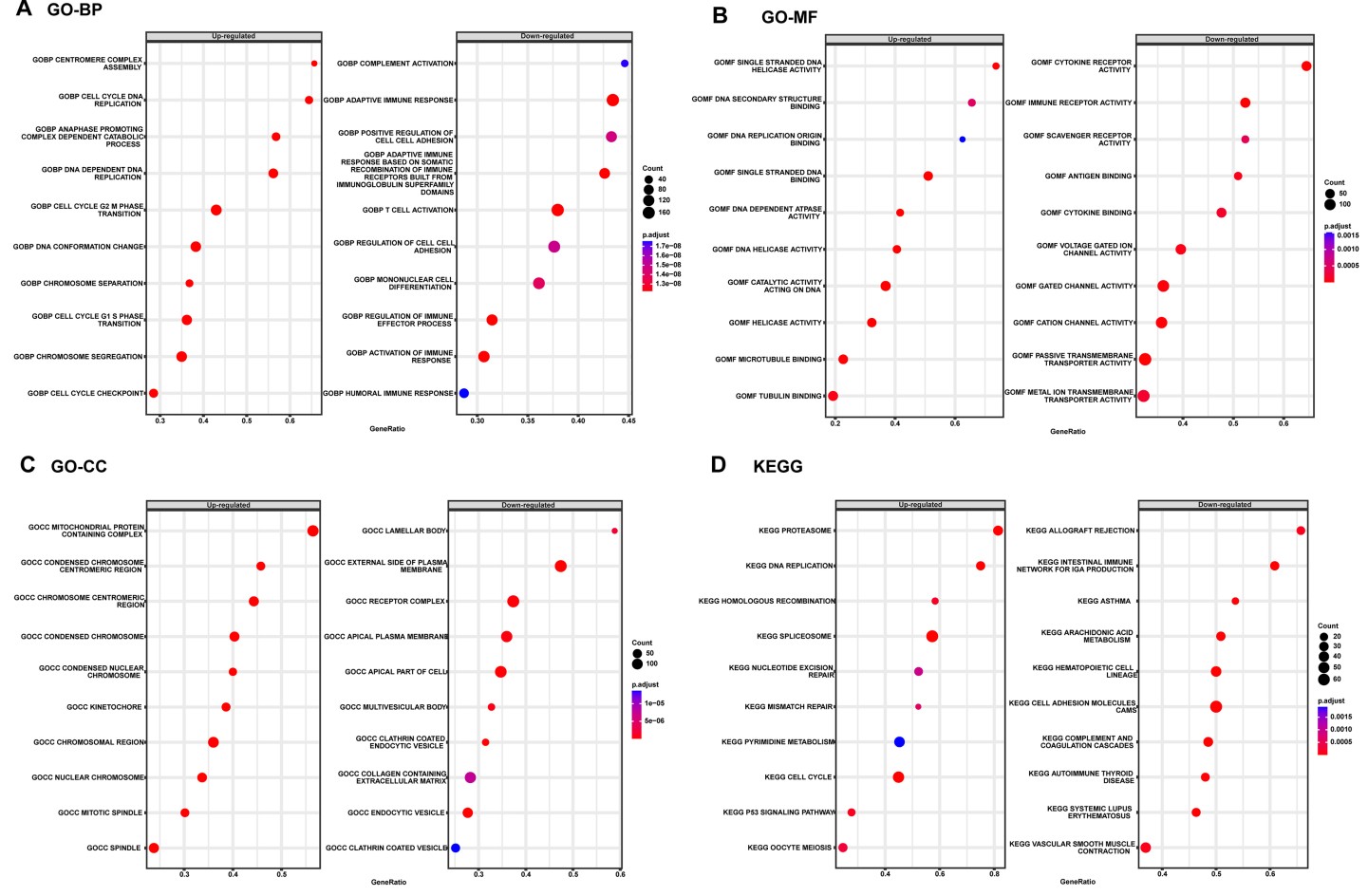

**Figure 4  The GSEA enrichment analysis in TCGA cohort.** (A) GO BP analysis of the autophagy score. (B) GO MF analysis of the autophagy score. (C) GO CC analysis of the autophagy score. (D) KEGG pathway enrichment of the autophagy score. KEGG, Kyoto Encyclopedia of Genes and Genomes; CC, Cellular Component; MF, Molecular Function; BP, Biological Process; GO, GeneOntology; GSEA, Gene Set Enrichment Analysis.

containing complex, condensed chromosome and chromosomal region, whereas the top three down-regulated GO CC terms were external side of plasma membrane, apical plasma membrane and receptor complex. Moreover, the KEGG pathway enrichment showed this meaningful autophagy score was involved in spliceosome, cell cycle, cell adhesion molecules cams and p53 signaling pathway.

## The correlation between the autophagy score and clinical features

In our study, the relationship between the autophagy score and clinical features was further analyzed. The autophagy score was not related to age, gender and smoking history (all $P > 0.05$, Fig. S1). We found that there was no relationship between the autophagy score and EGFR mutation status ($P = 0.73$). Interestingly, wild type KRAS patients had higher autophagy scores ($P = 0.017$), while wild type TP53 patients had lower autophagy scores ($P = 6.4e-11$). In addition, we constructed differential analysis between normal lung tissue and LUAD tumor tissue in the TCGA LAUD cohort and the Second Xiangya hospital. We

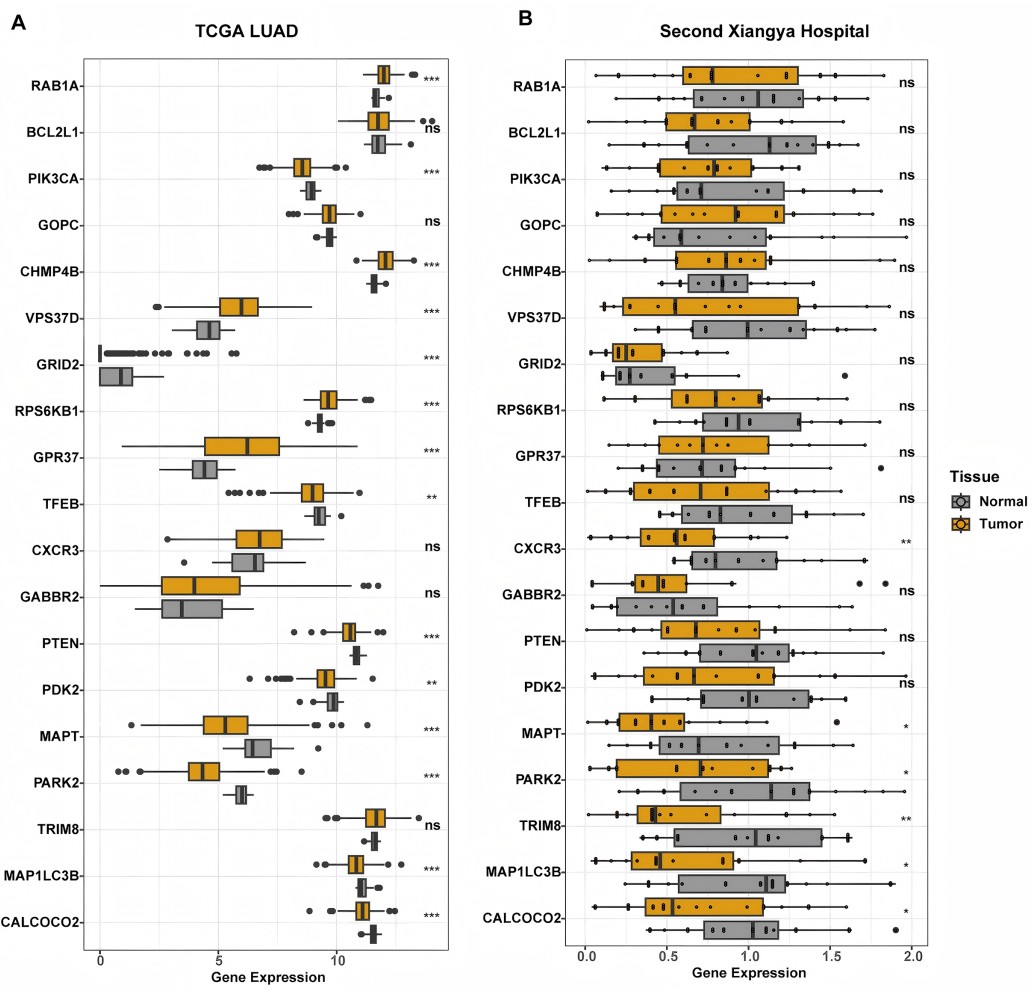

**Figure 5 Differential expression analysis of genes constructed the autophagy score.** (A) In TCGA cohort, RAB1A, CHMP4B, VPS37D, RPS6KB1, and GPR37 were significantly increased in tumor tissue. PIK3CA, GRID2, TFEB, PTEN, PDK2, MAPT, PARK2, MAP1LC3B, and CALCOCO2 significantly down-regulated in tumor tissue. BCL2L1, GOPC, CXCR3, GABBR2, and TRIM8 showed similar expression level between tumor and normal tissue. (B) In LUAD patients from the Second Xiangya hospital, CXCR3, MAPT, PARK2, TRIM8, MAP1LC3B, and CALCOCO2 showed lower expression in tumor tissue. While other genes had similar expression level between normal and tumor tissue. **$P < 0.01$; ***$P < 0.001$.

found that protective genes MAPT, MAP1LC3B, and CALCOCO2 were significantly down-regulated in both cohort (Fig. 5).

## The autophagy score was related to immune microenvironment

To deeply comprehend the potential contact between immune microenvironment and recurrence risk, we explored their relationship using a series of analytical methods. First, we analyzed the association between the autophagy score and different immune cells. In this study, the results showed the autophagy score was positively related to pro B cell, Th1

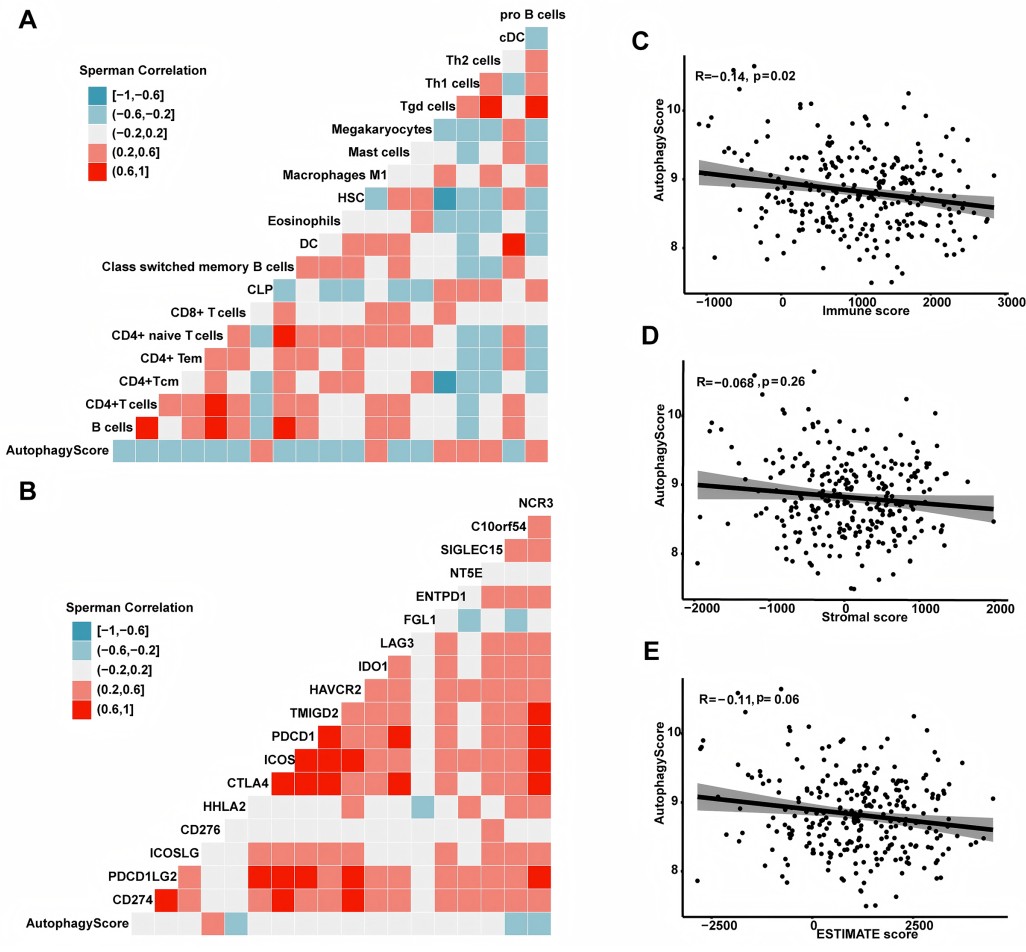

**Figure 6 The association between autophagy score and immune microenvironment.** (A) The relationship with immune cells. (B) The relationship with immunocheckpoint molecules. (C) The autophagy score was associated with the immune score (r = −0.14, $P$ = 0.02). (D) The autophagy score had no relationship with the stromal score (r = −0.068, $P$ = 0.26). (E) The autophagy score had a weak correlation with the ESTIMATE score (r = −0.11, $P$ = 0.06).

cell, Th2 cell and macrophage M1 cell, but negatively correlated with DC cell, mast cell, B cell, CD4+ and CD8+ T cell (all $P < 0.05$, Fig. 6). In addition, we also examined the association between the autophagy score and immune checkpoint molecules, such as CD274, PDCD1LG2, ICOSLG, CD276, HHLA2, CTLA4, *etc*. In general, the expression of CD276 was higher in high-risk group, while HHLA2, C10orf54 and NCR3 were higher in low-risk group (all $P < 0.05$). Next, we calculated the immune score by ESTIMATE algorithm and found that the autophagy score was negatively related to immune score (r = −0.14, $P$ = 0.02). In addition, the autophagy score was negatively correlated with ESTIMATE score, but the statistical difference was weak (r = −0.11, $P$ = 0.06). There seemed to be a correlation between the autophagy score and stromal score, but no statistical significance was observed (r = −0.068, $P$ = 0.26).

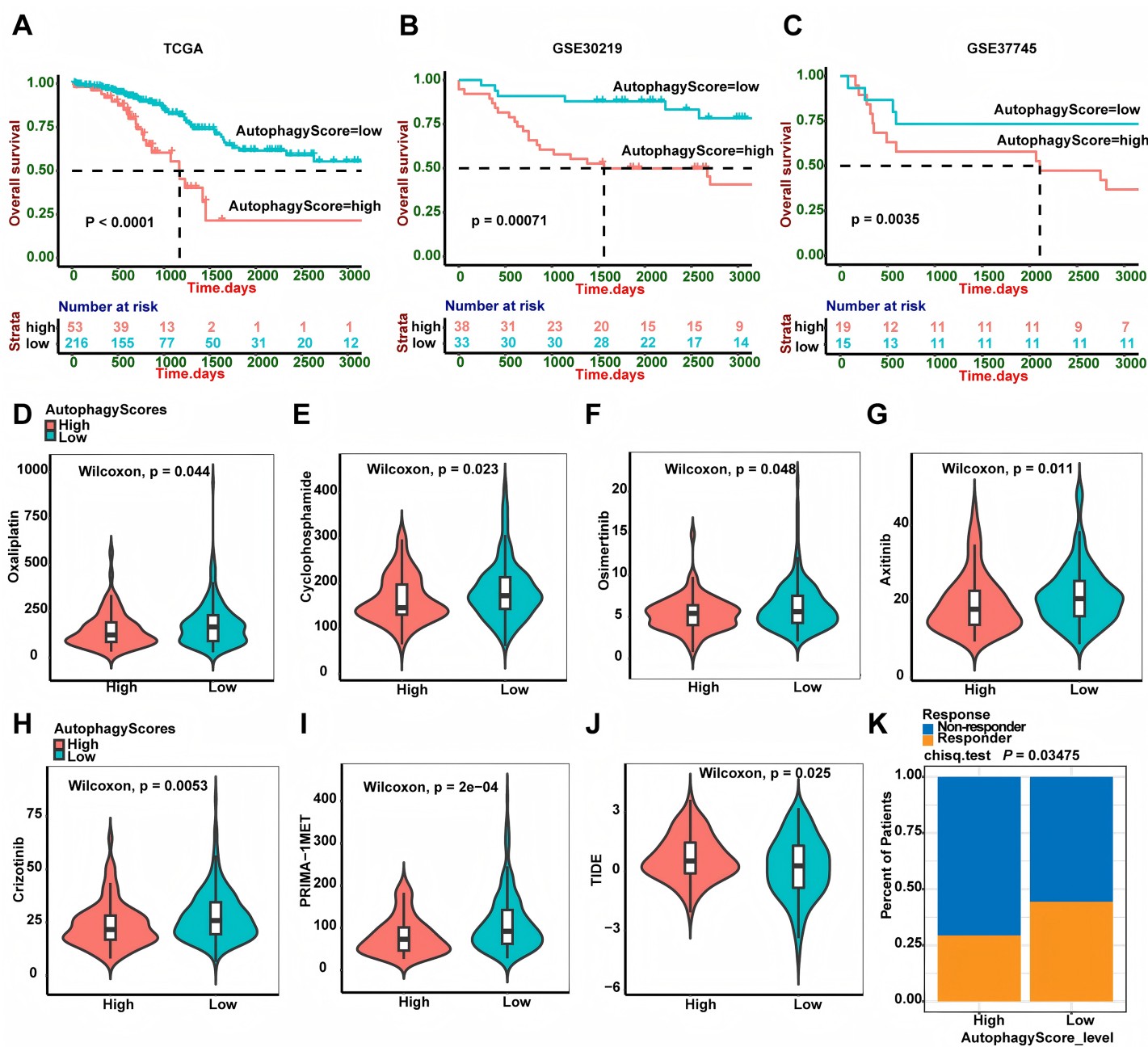

**Figure 7 Autophagy score predicted OS and treatment efficacy of chemotherapy, targeted therapy and immunotherapy.** (A–C) Overall survival analysis of the autophagy score in TCGA ($P < 0.0001$), GSE30219 ($P = 0.00071$) and GSE3774 ($P = 0.0035$) cohorts. (D–I) The estimated IC50 of oxaliplatin ($P = 0.044$), cyclophosphamide ($P = 0.023$), osimertinib ($P = 0.048$), axitinib ($P = 0.011$), crizotinib ($P = 0.0053$) and PRIMA-1MET ($P = 2e-04$) between two groups. (J) The relationship between the TIDE score and the autophagy score ($P = 0.025$). (K) Immunotherapy responder rate in two groups ($P = 0.03475$). TIDE, Tumor Immune Dysfunction and Exclusion; OS, overall survival.

### The autophagy score predicted overall survival and treatment efficacy of chemotherapy, targeted therapy and immunotherapy

In this study, autophagy signature could well predict the recurrence of LUAD patients. In addition, we also wanted to further explore whether it could predict the patients' overall survival. Interestingly, high-risk patients had shorter OS time than low-risk patients both in the TCGA, GSE30219 and GSE37745 cohorts (Fig. 7, $P < 0.0001$, $P = 0.00071$ and $P = 0.0035$, respectively). Chemotherapy and targeted therapy are the most common methods to treat LUAD patients. We used the GDSC database to explore the potential response of some chemotherapy and targeted therapy drugs between different groups. It was worth noting that the IC50 of oxaliplatin and cyclophosphamide was lower in high-risk group patients, suggesting that patients in high risk group were more sensitive to these chemotherapy drugs. In addition, we also found that high-risk group patients had lower IC50 for osimertinib (EGFR-TKI), axitinib (VEGFR-TKI), crizotinib (EML4-ALK fusion inhibitor) and PRIMA-1MET (mutant p53 activator), suggesting that compared to low-risk group patients, high-risk group patients were sensitive to these targeted therapy drugs. Recently, immunotherapy is emerging as an important treatment for LUAD patients. We calculated the TIDE score to predict the immunotherapy response of the two groups of patients. Intriguingly, the result showed low-risk group patients had lower TIDE score, which suggested that these patients had better immunotherapy response.

## DISCUSSION

For a long time, people have realized that the prognosis of metastatic cancers is poor, and various tumors will eventually metastasize, which has become the basis for tumor staging (*Cserni et al., 2018*). LUAD, as the most frequent histological type of lung cancer, can also be retraced to this progression (*Perez-Johnston et al., 2022*; *Xia et al., 2022*; *Cserni et al., 2018*). Due to the popularity of low-dose computed tomography (LDCT) screening, the diagnostic rate of stage I patients has been significantly improved, but these patients may experience early recurrence. Therefore, it is particularly important to timely identify patients who are prone to relapse and give them personalized treatment strategies. Chemotherapy, targeted therapy and immunotherapy are common treatments for LUAD patients, so it is also critical to identify patients who respond best to these treatments. Autophagy is a way of cellular catabolism, which degrades potentially harmful substances by transporting them to lysosomes. And there are three main types, known as chaperone-mediated autophagy, microautophagy and macroautophagy. Its key role in protecting the homeostasis of the intracellular environment is well known (*Russell & Guan, 2022*; *Mizushima et al., 2008*). Autophagy is a double-edged sword for tumors, because it can not only promote the survival of tumor cells through providing nutrients, but also prevent the formation of tumors (*Zada et al., 2021*; *Chen et al., 2019*). In recent years, more and more evidences have shown that autophagy was the key mechanism of tumor occurrence and drug resistance, and also the key factor of recurrence and metastasis (*Auberger & Puissant, 2017*; *Hu et al., 2021*; *Smith & Macleod, 2019*; *Limagne et al., 2022*; *Vera-Ramirez et al., 2018*; *Ren et al., 2022*). Some studies have focused on autophagy-related prognostic indices in different cancers (*Li et al., 2022*; *Liu et al., 2019*).

However, whether autophagy-related signature may become the biomarkers of tumor relapse and therapeutic response in stage I LUAD patients is still a knowledge gap. Therefore, in this current research, we established an autophagy gene related risk model to predict the recurrence and treatment response.

Compared to traditional biostatistical methods, machine learning (LASSO and Cox regression analysis are the most common) has great advantages in analyzing large sample datasets (*Ngiam & Khor, 2019*). In this research, we identified 19 autophagy related hub genes by LASSO analysis and created a risk model to calculate the autophagy score. Cox regression indicated the autophagy score was an independent risk factor for relapse of LUAD patients with stage I. The external validation of GEO dataset further proved the accuracy of the autophagy score model. This was the first time to establish an autophagy related risk model that could predict the relapse risk of LUAD patients with stage I. In addition, we further investigated the overall survival prognostic value of this risk model. We found it could also well predict LUAD patients' overall survival and high autophagy score patients had poor OS. To further promote the clinical application of this autophagy score model, we developed a nomograph to predict the RFS of LUAD patients by combing the autophagy score, age, gender and smoking history. Of these 19 hub genes, nine were risk factors (RAB1A, BCL2L1, PIK3CA, GOPC, CHMP4B, VPS37D, GRID2, RPS6KB1 and GRP37), and 10 were protective factors (CALCOCO2, MAP1LC3B, TRIM8, PARK2, MAPT, PDK2, PTEN, GABBR2, CXCR3 and TFEB). Among them, MAP1LC3B (microtubule associated protein 1 light chain 3 beta) was the core gene, which is one of the most famous autophagy related proteins (*Racanelli et al., 2018*). When macroautophagy is activated, MAP1LC3B, coupled with phosphatidyl-ethanolamine, targets to autophagic membranes (*Mizumura et al., 2012*). It has been shown that MAP1LC3B-II is positively related to ferroptosis sensitivity in ovarian cancer cells (*Li et al., 2021*). Moreover, overexpression of MAP1LC3B *in vitro* can prevent the development of Hermansky-Pudlak syndrome correlated with pulmonary fibrosis (*Ahuja et al., 2016*). Interestingly, MAP1LC3B has different roles in prognosis and clinicopathological features of distinct cancers (*Liu et al., 2018*). In this study, MAP1LC3B was a protective factor for the recurrence of stage I LUAD. MAP1LC3B exhibited the highest weight coefficient in PPI network. In addition, MAP1LC3B was down-regulated in tumor tissue both in TCGA and local cohort. These results suggested that MAP1LC3B may function as a crucial autophagy regulating and tumor development in LUAD.

PTEN (phosphatase and tensin homolog) is known as a tumor suppressor, which is mutated with a high frequency in human cancer (*Song, Salmena & Pandolfi, 2012*). PTEN tumor-suppressor activity mainly depends on its ability to inhibit the activation of PI3K/AKT, which controls various biological processes, including cell proliferation, migration and metabolism (*Song, Salmena & Pandolfi, 2012*; *Worby & Dixon, 2014*). PTEN promotes the induction of autophagy by facilitating the production of LC3-II (*Boosani, Gunasekar & Agrawal, 2019*). Studies showed that PTEN nuclear translocation promotes autophagy of cancer cells in response to DNA-damaging agents (*Chen et al., 2015*). BCL2L1, also known as Bcl-extra (Bcl-x), is an important member of the B-cell lymphoma 2 (Bcl-2) family and regulates cell fate (*Dou et al., 2021*). It has two antagonistic isoforms, Bcl-xL (blocking

apoptosis) and Bcl-xS (promoting apoptosis) (*Boise et al., 1993*). Apart from apoptosis, Bcl-xL is also considered to be involved in autophagy (*Zhang et al., 2018*). Specifically, Bcl-xL is identified to inhibit class III PI3K pathway-mediated macroautophagy (*Li, He & Ma, 2020*; *Zhou, Yang & Xing, 2011*). Recent studies have shown ectopic expression of Bcl-x isoforms is related to a variety of hallmarks of cancers (*Keitel et al., 2014*; *Li et al., 2020*). In the current study, BCL2L1 was identified as a risk factor for the recurrence of stage I LUAD. PARK2 encodes Parkin and is a pathogenic gene for cancers and neurodegenerative diseases (*Zhang et al., 2020*). Under stress conditions, Parkin can transfer to damaged mitochondria, promote mitochondrial proteins ubiquitination, and trigger mitophagy (*Harper, Ordureau & Heo, 2018*). Interestingly, recent studies have shown that Parkin can also act as a tumor suppressor, and somatic and germ line mutations in PARK2 is linked to multiple human cancers, including lung cancer (*Zhang et al., 2020*). RAB1A encodes GTPases and has been identified as mediating vesicular transport between the Golgi apparatus and endoplasmic reticulum (ER) (*Hutagalung & Novick, 2011*). In addition to its role in vesicular trafficking, Rab1A protein also has other functions, including cell migration, nutrient sensing and autophagy regulation (*Zoppino et al., 2010*; *Thomas et al., 2014*; *Wang et al., 2010*). Rab1A overexpression is related to poor prognosis and activates the mTORC1 pathway to promote tumor progression in hepatocellular carcinoma and colorectal cancer (*Thomas et al., 2014*).

CALCOCO2 (calcium binding and coiled-coil domain 2), a well-known xenophagy receptor, has been shown to regulate autophagosome maturation containing pathogen. And it may play a role in the organization of actin cytoskeleton and ruffle formation (*Morriswood et al., 2007*; *Boyle, Ravenhill & Randow, 2019*; *Cui et al., 2021*). Notably, many researches have shown CALCOCO2 may be correlated with cancer progression through interacting with tumor-associated signaling pathways such as NF-κB signaling pathway (*Leymarie et al., 2017*). In this study, CALCOCO2 showed the highest coefficient in protective factor of autophagy score. Differential analysis also reveled that CALCOCO2 significantly decreased in tumor tissue, suggesting that CALCOCO2 has the potential value as a predictive factor.

TRIM8 (tripartite motif containing 8) is identified as an E3 ubiquitin ligase protein, also named as GERP (glioblastoma expressed RING-finger protein). Recent researches showed that TRIM8 plays a dual role as an oncogene and a tumor suppressor gene, and produces a marked effect in the interaction between innate immunity and cancer (*Bhaduri & Merla, 2020*). In addition, under genotoxic stress conditions, TRIM8 has been reported to enhance autophagy flow through lysosomal biogenesis, thus degrading the cleaved Caspase-3 subunit and promoting cancer cell survival (*Roy et al., 2018*). CHMP4B (charged multivesicular body protein 4B) is a subunit of the ESCRT (endosomal sorting complex required for transport)-III complex, which produces a marked effect in mitotic cell division and the abscission of cytokinetic membrane (*Wollert et al., 2009*; *Elia et al., 2011*). A recent study reported that CHMP4B could be involved in autophagolysosomal degradation of micronuclei (*Sagona, Nezis & Stenmark, 2014*). CHMP4B with Vps4A mediates beta-catenin localization and exosome release to inhibit EMT (epithelial-mesenchymal transition) in liver cancer (*Han et al., 2019*). PDK2 (pyruvate dehydrogenase

kinase 2) is related to mitochondrial metabolism and its overexpression may play a key role in both cancer and diabetes (*Kitamura et al., 2021*). It has been shown that PDK2 regulates the PINK1/PARKIN-mediated mitophagy by modulating PARL β cleavage (*Shi & McQuibban, 2017*).

Next, we further investigated the molecular features of different risk groups by GSEA. GO and KEGG enrichment analysis showed significant enrichment of cell cycle, immunity and cell adhesion, including cell cycle checkpoint, adaptive immune response, T cell activation, activation of immune response, immune receptor activity and p53 signaling pathway. In addition, we further explored the association between the autophagy score and clinical characteristics, as well as driver gene mutation. Although the autophagy score was not correlated with clinical characteristics, it was significantly correlated with TP53 and KRAS gene mutation status. KRAS (kirsten rat sarcoma viral oncogene homolog) encodes a protein, which belongs to the small GTPase superfamily and its mutation is a genetic driver of NSCLC and many other cancers (*Zhu et al., 2022*). The activation of KRAS activates multiple downstream pathway (such as PI3K and MAPK), which is correlated with tumorigenesis and poor prognosis (*Zhu et al., 2022*; *Drosten et al., 2010*). In the present research, we found KRAS mutation patients had a lower autophagy score. TP53, a tumor suppressor gene, is found to have mutation in 50% cancers. Loss of TP53 function not only leads to tumor progression, but also affects the response to anticancer drugs, especially those that cause DNA damage (*Wang, Strasser & Kelly, 2022*). We found that TP53 mutation patients had a higher autophagy score, which provides some clues to further study the relationship between autophagy and TP53 gene mutation.

Tumor immune microenvironment (TIME) is quite complex and consists of tumor cells, tumor infiltrating immune cells and cytokines. Tumor infiltrating immune cells mainly contain dendritic cells (DC), M1-polarized macrophages, effector T cells (cytotoxic CD8+ T cells and effector CD4+ T cells), Tregs, MDSCs, M2-polarized macrophages, *etc*. More and more researches have indicated that tumor infiltrating immune cells play a pivotal role in the development and therapeutic response of many cancers (*Lv et al., 2022*; *Al-Shibli et al., 2008*). In the present study, we found the autophagy score was negatively related to DC, CD8+ and CD4+ T cells. Moreover, we discovered a positive relationship between the autophagy score and immune checkpoint molecule CD276. CD276, also named as B7-H3, belongs to B7 family and plays an immunosuppressive role in the tumor microenvironment (*Zhang et al., 2018*). Moreover, we also applied the ESTIMATE method to calculate the ESTIMATE score, stromal score and immune score. Interestingly, the autophagy score was negatively related to immune score, which was in accordance with the result of tumor infiltrating immune cells. Taken together, these findings not only supported the fact that high-risk patients had a higher probability of recurrence, but also suggested that they had a strong immunosuppressive microenvironment and might have poor effect on immunotherapy.

Additionally, we used the GDSC database to explore the potential response of patients to some targeted therapy and chemotherapy drugs. We found high-risk group patients had a lower IC50 for oxaliplatin and cyclophosphamide, suggesting that they might respond better to these chemotherapy drugs. Of note, high-risk group patients also had a lower

IC50 for mutant TP53 activator PRIMA-1MET, indicating a better response to PRIMA-1MET, which was consistent with the result of TP53 gene mutation. Also, high-risk group patients had a better response to crizotinib, which was an inhibitor for EML4-ALK fusion (*Park et al., 2022*). Intriguingly, high-risk group patients also had a better response to osimertinib (belonging to EGFR-TKI) and axitinib (belonging to VEGFR-TKI) than low-risk patients. Recently, immunotherapy is emerging as an important treatment for LUAD patients. Therefore, the efficacy of immunotherapy was evaluated by calculating the TIDE score. Unexpectedly, the result showed high-risk group patients had a worse response to immunotherapy.

By assessing autophagy-related markers, the model can help identify patients at higher risk, guiding more personalized and tailored treatment plans. This would enable clinicians to prioritize interventions and monitor patients more closely for disease progression. There are several limitations in this study, primarily due to its retrospective design. Retrospective studies rely on previously collected data, which limits our ability to establish causal relationships. We can only infer correlations between variables rather than confirm causality. Another limitation is the potential for selection bias. Since we relied on specific inclusion criteria, and there may be risks of data omissions or biases towards certain groups, our sample may not fully represent the target population.

## CONCLUSION

In summary, we established an autophagy gene related risk model for predicting relapse of stage I LUAD patients and further demonstrated its accuracy through external validation cohorts. In addition, we also found its significance in predicting overall survival and preliminarily explored its potential molecular mechanisms. More importantly, we found that this risk model could provide guidance for chemotherapy, targeted therapy and immunotherapy, thereby bringing clinical benefits to LUAD patients.

## ACKNOWLEDGEMENTS

We acknowledge GEO and TCGA databases for providing us with valuable data.

### Funding

The work was supported by grants from the National Natural Science Foundation of China (Nos: 82272722, 82102805 and 82200019), and the Natural Science Foundation of Hunan Province (No: 2025JJ60603). The funders had no role in study design, data collection and analysis, decision to publish, or preparation of the manuscript.

### Grant Disclosures

The following grant information was disclosed by the authors:
National Natural Science Foundation of China: 82272722, 82102805 and 82200019.
Natural Science Foundation of Hunan Province: 2025JJ60603.

## Competing Interests

The authors declare that they have no competing interests.

## Author Contributions

- Hongmei Zheng conceived and designed the experiments, performed the experiments, analyzed the data, prepared figures and/or tables, authored or reviewed drafts of the article, and approved the final draft.
- Songqing Fan analyzed the data, authored or reviewed drafts of the article, and approved the final draft.
- Hongjing Zang analyzed the data, prepared figures and/or tables, and approved the final draft.
- Jiadi Luo analyzed the data, prepared figures and/or tables, and approved the final draft.
- Long Shu conceived and designed the experiments, performed the experiments, authored or reviewed drafts of the article, and approved the final draft.
- Jinwu Peng conceived and designed the experiments, authored or reviewed drafts of the article, and approved the final draft.

## Human Ethics

The following information was supplied relating to ethical approvals (*i.e.*, approving body and any reference numbers):

This study was reviewed and approved by the Ethics Review Committee of Central South University, Second Xiangya Hospital (Ethical Application Ref:LYEC2024-0253).

## Data Availability

The data is available at NCBI GEO: GSE30219 and GSE37745.

zheng, hongmei (2024). GSE30219_series_matrix.txt.gz. figshare. Dataset. https://doi.org/10.6084/m9.figshare.27633822.v1.

zheng, hongmei (2024). GSE37745_series_matrix.txt.gz. figshare. Dataset. https://doi.org/10.6084/m9.figshare.27633828.v1.

The TCGA data is available at the Cancer Genomics Browser of The University of California Santa Cruz (UCSC): http://xena.ucsc.edu.

All autophagy-related genes are available at the human autophagy moderator database: HAMdb, http://hamdb.scbdd.com.

## Supplemental Information

Supplemental information for this article can be found online at http://dx.doi.org/10.7717/peerj.19366#supplemental-information.

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
