# Peer review of "A comprehensive analysis identified an autophagy-related risk model for predicting recurrence and immunotherapy response in stage I lung adenocarcinoma"

_PeerJ, doi:10.7717/peerj.19366_

## Round 0.1 · original submission · Major Revisions

Two experts in the field reviewed your manuscript. As you can see from their comments below, both give relatively positive comments on this work. Please read the comments carefully and revise the manuscript accordingly. I believe that the revision process will not be so heavy.

Reviewer 1 ·

Basic reporting

1. Overall the English writing is OK. But there are some places where the language could be improved. Line 166-170 when you brought up GO terms, please use “the proteins that are involved in these pathways.” Line 193- “analysis was extended” need to be changed. I suggest you clearly demonstrate what exactly you are trying to convey to the audience.

2. Intro and background are well referenced.

3. The overall structure conforms to PeerJ standard; The figures are corrected labeled; however, some panels were not mentioned in the main text. For instance, Figure 6d was not in the main text. Please also conjecture why three scores are all negatively related to your autophagy score.

4. Please explain to general audience why the value of the AUC indicated good performance of the risk model in line 160.

Experimental design

The manuscript analyzed autophagy genes that related to the recurrence of LUAD patients, they created and validated a risk model based on the analysis. They also correlated the autophagy score to biological pathways, clinical features, and immune microenvironment. At last, they showed that the autophagy score can be used to predict survival and treatment efficacy of therapies.

Overall, the research gives a meaningful correlation between autophagy and LUAD. The question is meaningful as well. The data is valid. Method can be replicated based on their description.

I have only one concern about the background of the scientific question they are dealing with:

Li et al (Front Immunol 2022, citation 12) have shown that the autophagy signatures can be used for therapy efficiency prediction in LUAD patients. Please explain why the autophagy prediction on Stage I LUAD patients is important to get more justification for your study. Accordingly, please tone down the description of Lines 236-239.

Validity of the findings

The data is valid, well-statistically analyzed. The conclusion are well stated.

·

Basic reporting

• Clarity and Language: The manuscript is written in clear, professional English. The language is unambiguous and suitable for an international audience.
• Introduction and Background: The introduction provides a thorough background on LUAD, autophagy, and the current knowledge gap regarding the prediction of relapse and therapeutic response in stage I LUAD patients. The literature is well-referenced and relevant.
• Structure: The structure conforms to PeerJ standards and is well-organized, with clear sections and sub-sections.
• Figures and Tables: The figures are relevant, high quality, and well-labeled. The tables provide necessary information, such as clinical characteristics and primer sequences.
• Raw Data: The authors have provided raw data, which is accessible through the TCGA and GEO databases.

Experimental design

• Originality and Scope: The research is original and falls within the scope of the journal. The study addresses a significant knowledge gap by focusing on autophagy-related genes in stage I LUAD.
• Research Question: The research question is well-defined, relevant, and meaningful. The study aims to construct a risk model based on autophagy-related genes to predict recurrence and therapeutic response in stage I LUAD patients.
• Methodology: The methods are described in sufficient detail, allowing for replication. The use of LASSO regression, real-time PCR, PPI network analysis, GSEA, and ESTIMATE algorithm are appropriate for the study's objectives.
• Ethical Standards: The study meets high technical and ethical standards, with ethical approval obtained from the relevant committee.

Validity of the findings

Please see my comments below.• Data Robustness: The underlying data are robust, statistically sound, and controlled. The authors have used multiple validation cohorts (TCGA and GEO datasets) to ensure the reliability of their findings.
• Statistical Analysis: The statistical analysis is appropriate, with multivariate Cox regression, ROC curve analysis, and decision curve analysis used to validate the risk model.
Conclusions: The conclusions are well-stated and linked to the original research question. The authors have limited their conclusions to the supporting results, avoiding overinterpretation.

Additional comments

• Strengths:
o The study is comprehensive, utilizing multiple databases and bioinformatics tools to construct and validate the autophagy-related risk model.
o The inclusion of external validation cohorts (GSE30219 and GSE37745) strengthens the findings.
o The study not only predicts recurrence but also explores the immune microenvironment and therapeutic response, providing a holistic view of the role of autophagy in LUAD.
• Weaknesses:
o The study could benefit from a more detailed discussion on the limitations, such as potential biases in the datasets used or the generalizability of the findings to other populations.
o While the authors mention the potential clinical application of the risk model, further discussion on how this model could be integrated into clinical practice would be valuable.
5. Suggestions for Improvement
• Discussion of Limitations: The authors should discuss the limitations of their study, such as the retrospective nature of the data and the potential for selection bias.
• Clinical Implications: A more detailed discussion on how the autophagy-related risk model could be used in clinical practice, including potential challenges and future directions, would enhance the manuscript.
• Language and Grammar: While the language is generally clear, there are a few minor grammatical errors that could be corrected. For example, in the abstract, "We established autophagy score based on 19 autophagy genes" could be revised to "We established an autophagy score based on 19 autophagy genes."

---

## Round 0.2 · accepted · Accept

Since both original reviewers now recommend that your revised manuscript be accepted, I am happy to make the decision. Congratulations!

Reviewer 1 ·

Basic reporting

The author now address all my concerns.

Experimental design

The author now address all my concerns.

Validity of the findings

The author now address all my concerns.

·

Basic reporting

The authors addressed my prior commence and it is my pleasure to accept their work.

Experimental design

The authors addressed my prior commence and it is my pleasure to accept their work.

Validity of the findings

The authors addressed my prior commence and it is my pleasure to accept their work.

Additional comments

The authors addressed my prior commence and it is my pleasure to accept their work.